# Activation of cryptic splicing in bovine *WDR19* is associated with reduced semen quality and male fertility

Maya Hiltpold[1], Guanglin Niu[2], Naveen Kumar Kadri[1], Danang Crysnanto[1], Zih-Hua Fang[1], Mirjam Spengeler[3], Fritz Schmitz-Hsu[4], Christian Fuerst[5], Hermann Schwarzenbacher[5], Franz R. Seefried[3], Frauke Seehusen[6], Ulrich Witschi[4], Angelika Schnieke[2], Ruedi Fries[7], Heinrich Bollwein[8], Krzysztof Flisikowski[2], Hubert Pausch[1] *

1 Animal Genomics, ETH Zürich, Lindau, Switzerland, 2 Livestock Biotechnology, TU München, Freising, Germany, 3 QualitasAG, Zug, Switzerland, 4 Swissgenetics, Zollikofen, Switzerland, 5 ZuchtData, Wien, Austria, 6 Institute of Veterinary Pathology, University of Zurich, Zurich, Switzerland, 7 Animal Breeding, TU München, Freising, Germany, 8 Clinic of Reproductive Medicine, University of Zurich, Zürich, Switzerland

* hubert.pausch@usys.ethz.ch

**Data Availability Statement:** Whole-genome sequencing data of 42 Brown Swiss bulls are available at the European Nucleotide Archive (ENA) of EMBL-EBI under sample accession numbers

## Abstract

Cattle are ideally suited to investigate the genetics of male reproduction, because semen quality and fertility are recorded for all ejaculates of artificial insemination bulls. We analysed 26,090 ejaculates of 794 Brown Swiss bulls to assess ejaculate volume, sperm concentration, sperm motility, sperm head and tail anomalies and insemination success. The heritability of the six semen traits was between 0 and 0.26. Genome-wide association testing on 607,511 SNPs revealed a QTL on bovine chromosome 6 that was associated with sperm motility ($P = 2.5 \times 10^{-27}$), head ($P = 2.0 \times 10^{-44}$) and tail anomalies ($P = 7.2 \times 10^{-49}$) and insemination success ($P = 9.9 \times 10^{-13}$). The QTL harbors a recessive allele that compromises semen quality and male fertility. We replicated the effect of the QTL on fertility ($P = 7.1 \times 10^{-32}$) in an independent cohort of 2481 Brown Swiss bulls. The analysis of whole-genome sequencing data revealed that a synonymous variant (BTA6:58373887C>T, rs474302732) in *WDR19* encoding WD repeat-containing protein 19 was in linkage disequilibrium with the fertility-associated haplotype. WD repeat-containing protein 19 is a constituent of the intraflagellar transport complex that is essential for the physiological function of motile cilia and flagella. Bioinformatic and transcription analyses revealed that the BTA6:58373887 T-allele activates a cryptic exonic splice site that eliminates three evolutionarily conserved amino acids from WDR19. Western blot analysis demonstrated that the BTA6:58373887 T-allele decreases protein expression. We make the remarkable observation that, in spite of negative effects on semen quality and bull fertility, the BTA6:58373887 T-allele has a frequency of 24% in the Brown Swiss population. Our findings are the first to uncover a variant that is associated with quantitative variation in semen quality and male fertility in cattle.

listed in S2 Table. Accession numbers for 295 bulls that were used to assess allele frequency are available at the European Nucleotide Archive (ENA) of EMBL-EBI under sample accession numbers listed in S4 Table. RNA sequencing data from testicular tissue from a bull homozygous for the WDR19 mutation have been deposited at the Sequence Read Archive of the NCBI under study accession PRJNA616249 and sample accession SAMN14485268. Transcriptome data used to quantify transcript abundance in bovine testes are available at the ENA under sample accession numbers SAMN09205186-SAMN09205191. Haplotypes, eigenvectors and phenotypes of 794 Swiss BSW bulls that were used to detect the QTL at BTA6 are available as S1 Data. The R script to carry out the haplotype-based association study is available as S2 Data.

**Funding:** HP and HB received funding from Swissgenetics, Zollikofen, Switzerland (https://swissgenetics.ch/); Grant-ID: 13453. HP received funding from the Förderverein Biotechnologieforschung e.V. (FBF), Bonn, Germany (https://www.fbf-forschung.de/); Grant-ID: 2-70024-18. HP received funding from the Swiss Federal Office for Agriculture, Bern (https://www.blw.admin.ch/blw/de/home.html); Grant-ID: 927000958. The funders had no role in study design, data collection and analysis, decision to publish, or preparation of the manuscript.

**Competing interests:** I have read the journal's policy and the authors of this manuscript have the following competing interests: FSH and UW are employees of Swissgenetics.

## Author summary

In cattle farming, artificial insemination is the most common method of breeding. To ensure high fertilization rates, ejaculate quality and insemination success are closely monitored in artificial insemination bulls. We analyse semen quality, insemination success and microarray-called genotypes at more than 600,000 genome-wide SNP markers of 794 bulls to identify a recessive allele that compromises semen quality. We take advantage of whole-genome sequencing to pinpoint a variant in the coding sequence of *WDR19* encoding WD repeat-containing protein 19 that activates a novel exonic splice site. Our results indicate that cryptic splicing in *WDR19* is associated with reduced male reproductive performance. This is the first report of a variant that contributes to quantitative variation in bovine semen quality.

## Introduction

Reproduction plays a pivotal role in dairy and beef production. Delayed conception compromises profit and may lead to the unintended culling of cows and heifers [1–3]. Most cows and heifers are bred using cryoconserved semen from artificial insemination (AI) bulls. Because each AI bull is mated to many cows, factors contributing to conception can be partitioned accurately for males and females using multiple trait animal models [4,5].

The fertility of AI bulls can be quantified using insemination success adjusted for environmental and genetic effects [4,6]. However, the heritability of bull fertility is low [6–8]. Semen traits that are routinely recorded at AI centers such as have higher heritability than bull fertility [9–12]. Traits that are routinely assessed from fresh ejaculates include semen volume, sperm concentration, sperm motility, and the proportion of sperm with head and tail anomalies. Computer-assisted and flow-cytometric sperm analyses sometimes complement the macroscopic and microscopic evaluations of semen samples [13].

Ejaculates that fulfill the quality requirements for AI [14] are diluted with cryoprotective semen extenders and filled in straws containing between 15 and 25 million spermatozoa per straw. In order to ensure uniform insemination success within and between bulls, the number of spermatozoa per straw is higher for ejaculates that fulfill the minimum requirements but contain some sperm with compensable defects [15].

Routinely recorded semen quality data facilitate investigation of the genetics of male fertility. SNP microarray-derived genotypes of AI bulls [16] can be imputed to the whole-genome sequence level using, e.g., the reference panel of the 1000 Bull Genomes Project consortium [17,18]. Large mapping cohorts with dense genotypes and semen traits provide high statistical power to detect male fertility-associated variants using genome-wide association testing.

Genome-wide association studies using microarray-derived genotypes provided evidence that inherited differences in semen quality are amenable to genome-wide association testing (e.g., [19–25]). However, low marker density resulted in large QTL confidence intervals that contained many genes and candidate causal variants. A missense variant (rs136195618) in the *PROP1* gene was postulated to be associated with bull fertility [26]. However, the effect of the missense variant on bull fertility was not validated in two independent populations [27]. Case-control association studies uncovered recessive variants in the *TMEM95*, *ARMC3* and *CCDC189* genes, that compromise semen quality and bull fertility in the homozygous state [28–30].

Here, we assess semen quality and fertility of 794 Brown Swiss (BSW) bulls using data from 26,090 routinely collected ejaculates. Genome-wide association testing revealed a QTL on bovine chromosome 6 that is associated with semen quality and fertility. Whole-genome

sequencing, transcription and protein analyses enabled us to detect a cryptic splice site variant in *WDR19* that is associated with reduced semen quality and fertility.

## Results

### Semen quality and fertility of BSW bulls

After applying rigorous quality control on the phenotype data including the removal of ejaculates that did not fulfill the requirements for AI, we considered 26,090 ejaculates of 794 BSW bulls for genetic analyses (Table 1). Bull fertility was estimated from female non-return rates at 56 days after the insemination for 591 bulls. Average values for volume, sperm concentration, sperm motility, sperm head and tail anomaly scores, and number of sperm per insemination straw were calculated using between 8 and 162 ejaculates (median: 20) per bull. The average ejaculate volume was 3.93 ml (Table 1). On average, each ejaculate contained $1.32 \times 10^9$ sperm per ml and 86% of the sperm were motile. Each ejaculate received scores between 0 and 3 indicating few and many sperm with head and tail anomalies, respectively. The vast majority of the ejaculates (93.4 and 87.6%) had very few sperm with head and tail anomalies. On average, each ejaculate was diluted into 319 straws that contained 16.45 million spermatozoa per straw.

Marked phenotypic correlations were detected between the semen traits (S1 Table). Phenotypic correlations were particularly high between head and tail anomalies (r = 0.84), tail anomalies and sperm count per straw (r = 0.68), and motility and tail anomalies (r = -0.75). Motility was positively correlated with bull fertility (r = 0.22). Sperm count per straw, head, and tail anomalies were negatively correlated (r ≤ -0.2) with bull fertility. The repeatability of the semen traits was between 0.38 and 0.61, but the heritability was clearly lower indicating that permanent environmental effects markedly affect semen quality (Table 1). The heritability was close to zero for the proportion of sperm with either head or tail anomalies. The heritability was 0.27, 0.25, 0.12 and 0.10 for ejaculate volume, sperm concentration, sperm count per straw, and motility, respectively, suggesting that genome-wide association testing between semen quality and dense genotypes in 794 bulls should have sufficient statistical power to detect QTL with moderate to large effects [31].

### A QTL for semen quality and fertility is located on bovine chromosome 6

To detect QTL for semen quality and fertility, we carried out genome-wide haplotype-based association tests that were based on additive and recessive modes of inheritance. We

**Table 1. Semen quality and fertility of 794 BSW bulls.**

|  | unit | min | mean (± sd) | max | repeatability | heritability |
|---|---|---|---|---|---|---|
| Ejaculates | n | 8 | 32.86 ± 28.69 | 162 |  |  |
| Volume | ml | 1.57 | 3.93 ± 0.94 | 7.31 | 0.41 ± 0.02 | 0.27 ± 0.05 |
| Concentration | $10^9$/ml | 0.47 | 1.32 ± 0.32 | 2.68 | 0.43 ± 0.02 | 0.25 ± 0.06 |
| Motility | % | 77.85 | 86.16 ± 2.33 | 92.27 | 0.44 ± 0.01 | 0.10 ± 0.04 |
| Sperm number per straw | mio | 15.00 | 16.45 ± 1.91 | 24.85 | 0.61 ± 0.01 | 0.12 ± 0.06 |
| Tail anomalies | score | 0.00 | 0.13 ± 0.26 | 1.53 | 0.43 ± 0.01 | 0.02 ± 0.03 |
| Head anomalies | score | 0.00 | 0.05 ± 0.14 | 0.90 | 0.38 ± 0.01 | 0[a] |
| Bull fertility | index | 65 | 100.6 ± 9.37 | 127 |  |  |
| Inseminations | n | 239 | 744 ± 1437 | 15690 |  |  |

Heritability ($h^2$) and repeatability were estimated using pedigree-based relationship coefficients and taking into account genetic, non-genetic, environmental and permanent environmental effects (see Material and Methods).

[a]The model for head anomalies converged at $10^{-9}$ as convergence criterion, for all other traits, the convergence criterion was $10^{-15}$.

considered six semen traits and male fertility (Table 1), respectively, of 794 and 591 BSW bulls that had partially imputed genotypes at 607,511 SNPs. In order to take population stratification and the resulting inflation of false-positive association signals into account, we included the top ten principal components of the genomic relationship matrix in the statistical models. The average genomic inflation factor lambda was $1.29 \pm 0.19$ and $1.12 \pm 0.07$ for the additive and recessive model, respectively, indicating that this corrective measure was mostly successful.

Haplotype-based association testing (additive model) revealed a QTL on chromosome 6 (S1 Fig) for sperm motility ($P = 4.1 \times 10^{-12}$), sperm head and tail anomalies ($P = 5.2 \times 10^{-16}$, $P = 1.3 \times 10^{-17}$), sperm count per straw ($P = 8.6 \times 10^{-12}$) and insemination success ($P = 1.5 \times 10^{-8}$). The strongest association signals resulted from three adjacent haplotypes with almost identical P values that were located between 57,335,668 and 57,993,128 bp. The QTL on chromosome 6 was not associated with ejaculate volume and sperm concentration. Only few haplotypes located on chromosomes other than BTA6 met the Bonferroni-corrected significance threshold for the seven traits analysed (S1 Fig).

The association of the QTL on chromosome 6 was more pronounced when the association tests were carried out assuming recessive inheritance (Fig 1). The strongest associations were detected for two overlapping haplotypes located between 57,538,068 and 57,993,128 bp. The P value of the top haplotype was $9.9 \times 10^{-13}$, $2.6 \times 10^{-26}$, $2.5 \times 10^{-27}$, $2.0 \times 10^{-44}$, and $7.2 \times 10^{-49}$ for bull fertility, sperm count per straw, motility, head anomalies, and tail anomalies, respectively. The P value of the top haplotype was higher (i.e., less significant) for bull fertility than semen quality. This is partly due to the fact that the number of bulls with fertility records was lower than the number of bulls with semen quality records. After correcting for multiple testing, the top haplotype was not associated with ejaculate volume ($P = 0.002$) and sperm concentration ($P = 0.65$). The frequency of the top haplotype was 24%. Of 794 AI bulls, 291 were heterozygous and 46 were homozygous for the top haplotype. The association signal on chromosome 6 was absent for all traits, when the association analysis was conditioned on the top haplotype indicating that the top haplotype fully accounts for the QTL.

Homozygosity for the top haplotype on chromosome 6 is associated with reduced sperm motility and fertility, increased sperm head and tail anomalies, and more sperm per straw (Fig 2). The semen quality and fertility of heterozygous bulls were normal corroborating recessive inheritance. Semen quality of homozygous bulls was only slightly reduced compared to either heterozygous or non-carrier bulls. For instance, with an average proportion of 83% motile sperm per ejaculate, homozygous bulls fulfill the minimum requirements for artificial insemination which is 70% [32]. However, more ejaculates were rejected for AI due to less than 70% motile sperm in homozygous than either heterozygous or non-carrier bulls (4.5% vs. 1.7%). Insemination straws of homozygous bulls contain 2.86 million additional sperm per straw. However, the fertility of homozygous bulls is reduced in spite of the compensation (Fig 2E).

## Homozygous haplotype carriers share a 2.38 Mb segment on BTA6

Haplotypes that were significantly associated with either semen quality or bull fertility were detected within an 15 Mb interval (between 50 and 65 Mb) on chromosome 6. One hundred and five haplotypes that were significantly associated with at least two of the seven traits studied were located between 55,348,382 and 65,408,468 bp (Fig 3A). We detected haplotypes that had markedly lower P values (e.g., $P < 1 \times 10^{-30}$ for sperm head and tail anomalies, $P < 1 \times 10^{-15}$ for motility) than surrounding haplotypes between 56,922,962 and 58,293,842 bp. A deeper analysis of the genotypes of 46 bulls that were homozygous for the top haplotype showed that they share a 2.38 Mb segment of extended autozygosity (between 57,465,157 and 59,846,532 bp) (Fig 3B). This segment contains 23 genes (*KLF3*, *TLR10*, *TLR6*, *FAM114A1*,

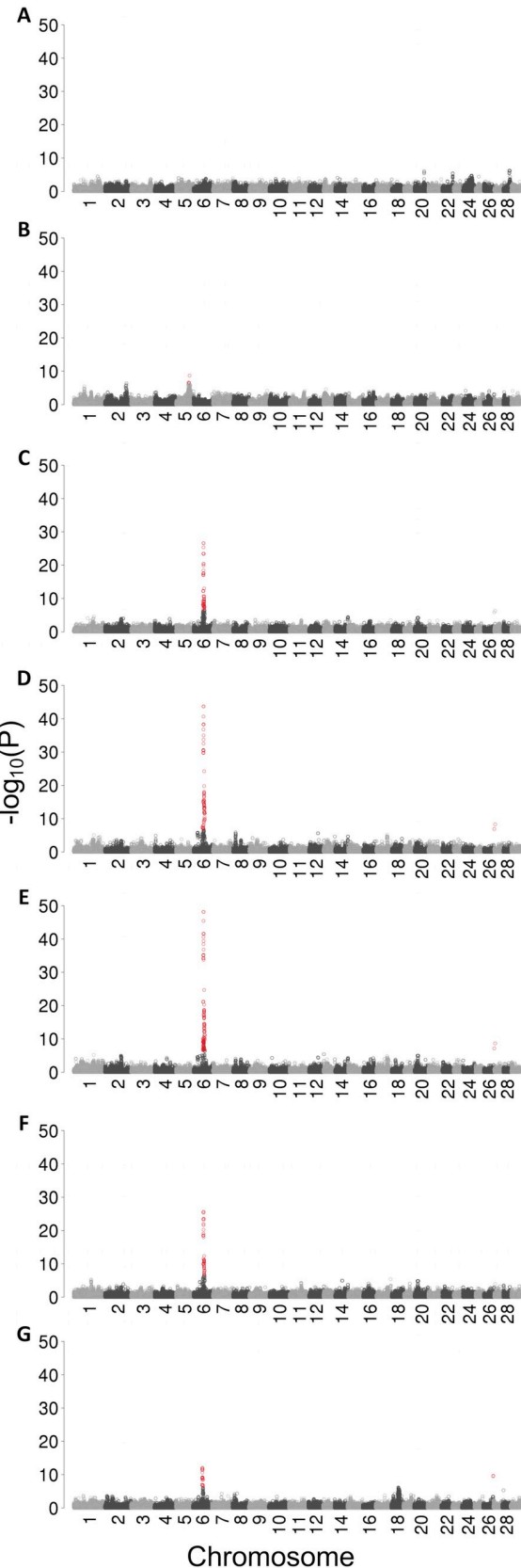

Fig 1. **Detection of QTL for semen quality and fertility in BSW bulls.** Manhattan plots representing the association (−log₁₀(P)) of haplotypes with (A) ejaculate volume (genomic inflation factor lambda = 1.25), (B) sperm concentration (lambda = 1.12), (C) sperm motility (lambda = 1.05), (D) sperm head anomalies (lambda = 1.10), (E) sperm tail anomalies (lambda = 1.12), (F) sperm per straw (lambda = 1.16), and (G) bull fertility (lambda = 1.06) assuming a recessive mode of inheritance. Red color indicates significantly associated haplotypes (P < Bonferroni-corrected significance threshold).

*ENSBTAG00000055220, TMEM156, KLHL5, WDR19, RFC1, KLB, RPL9, LIAS, UGDH, SMIM14, UBE2K, PDS5A, N4BP2, ENSBTAG00000049669, RHOH, CHRNA9, RBM47, NSUN7, APBB2*) (Fig 3C). Of these genes, 15 and 14 are expressed at medium to high levels (> 10 transcripts per million (TPM)) in the testes of newborn and mature males, respectively (S2 Fig) including *NSUN7* encoding NOP2/Sun RNA Methyltransferase Family Member 7. Loss-of-function alleles in *NSUN7* have been associated with low sperm motility and impaired male fertility in mice and humans [33],[34].

## A synonymous variant in *WDR19* is in linkage disequilibrium with the top haplotype

We analysed whole-genome sequencing data of 42 BSW animals for which the status for the BTA6 top haplotype was known from SNP microarray genotypes. Of the 42 BSW animals, 3

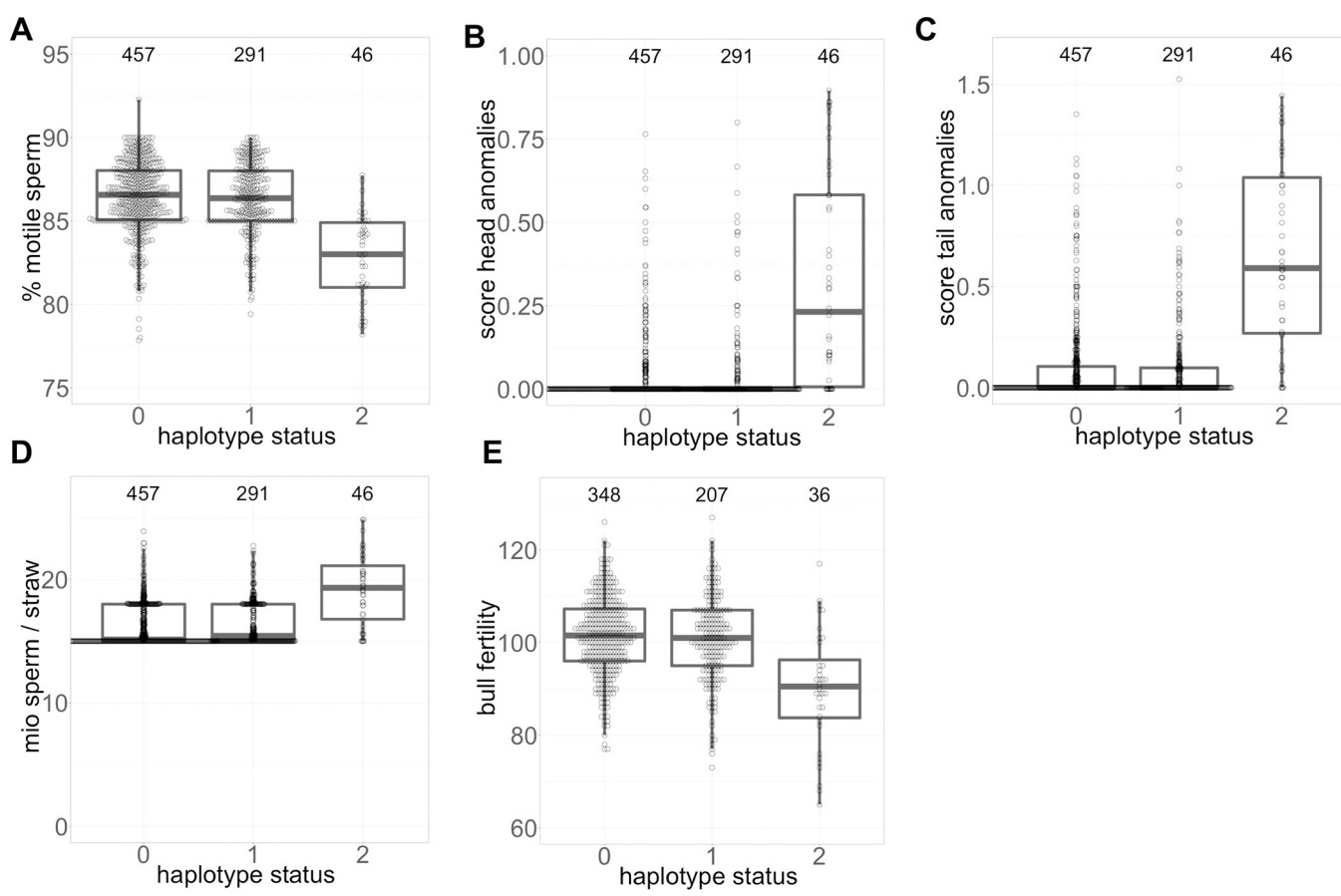

Fig 2. **A recessive haplotype compromises semen quality and bull fertility.** Boxplots representing the effect of the top haplotype on semen quality and fertility in non-carrier, heterozygous and homozygous bulls (haplotype status 0, 1, and 2, respectively). (A) Sperm motility of homozygous bulls is reduced by 1.6 phenotypic standard deviations ($\sigma_P$) (82.77 ± 2.59% vs. 86.37 ± 2.14%). Scores for sperm head (B) and tail (C) anomalies are increased by 2 and 2.1 $\sigma_P$ in homozygous bulls. (D) The reduced semen quality of homozygous bulls is compensated with an increased number of sperm per dose (19.15 ± 2.83 vs. 16.29 ± 1.7 million). (E) Fertility of homozygous bulls is reduced by 1.2 $\sigma_P$ (90.1 ± 12.3 vs. 101.27 ± 8.74).

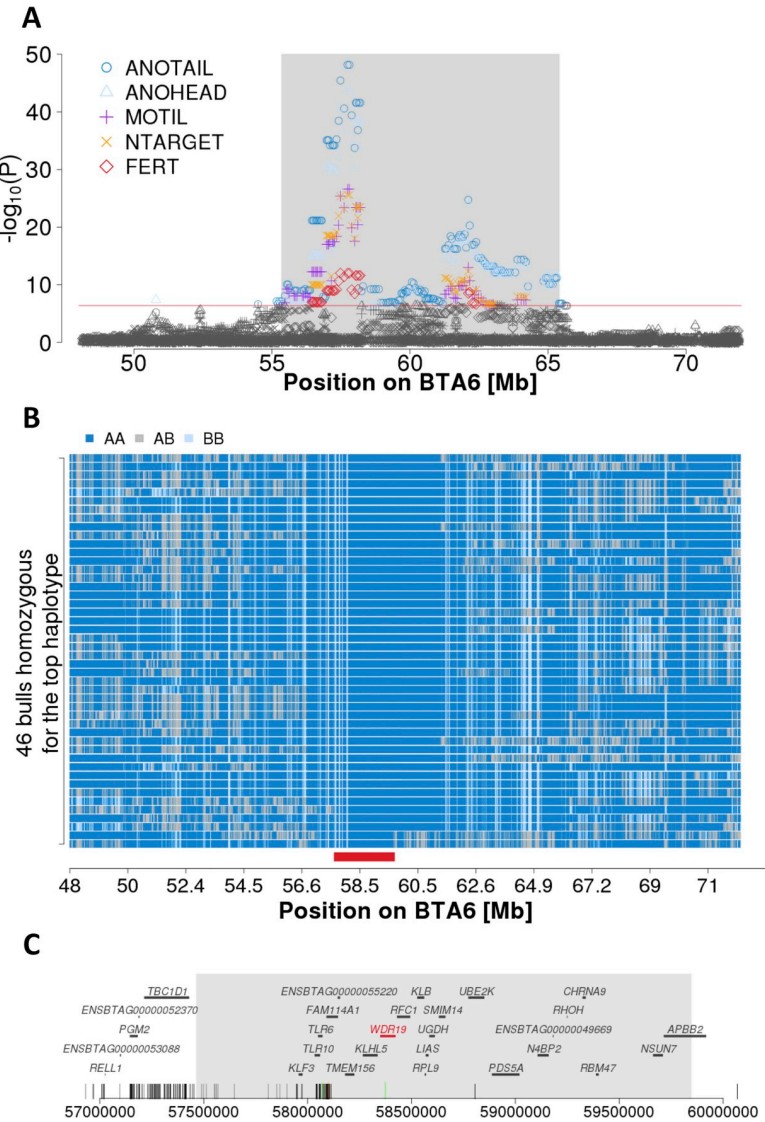

**Fig 3. Detailed view of the associated region on chromosome 6.** (A) Association (-log₁₀(P)) of haplotypes located between 48 and 72 Mb on BTA6 with head and tail anomalies (ANOHEAD, ANOTAIL), motility (MOTIL), number of sperm per straw (NTARGET) and bull fertility (FERT) assuming recessive inheritance. The red line represents the Bonferroni-corrected significance threshold. The grey shaded area contains haplotypes that were associated with at least two traits. (B) A segment of extended autozygosity was detected in 46 bulls that were homozygous for the top-haplotype. Blue and pale blue represent homozygous genotypes (AA and BB), heterozygous genotypes are displayed in grey. The solid red bar indicates the common 2.38 Mb segment of extended autozygosity (from 57,465,157 to 59,846,532 bp). (C) The segment of extended autozygosity encompasses 23 protein-coding genes. Vertical bars at the bottom represent variants that are compatible with recessive inheritance, including 126 that are located within the segment of extended autozygosity. Dark grey, light grey, green and orange bars represent intronic, intergenic, synonymous and missense variants, respectively.

were homozygous and 10 heterozygous carriers of the top haplotype. The remaining 29 bulls did not carry the top haplotype (S2 Table). The average sequencing depth of the 42 animals was 13.9-fold. Analysis of the sequencing read alignments and sequencing depth along chromosome 6 in three homozygous haplotype carriers using the *Integrative Genomics Viewer* [35] and *mosdepth* software [36] did not reveal large sequence variants that segregate with the fertility-associated haplotype.

As causal variants might reside outside of top haplotypes (e.g., [37]), we considered a 10 Mb interval (between 55 and 65 Mb, Fig 3A) that contained haplotypes that were associated with at least two of the traits studied to identify variants that were compatible with recessive inheritance. Within this interval, we detected 56,081 variants (47,686 SNPs and 8,395 Indels) that were polymorphic in the 42 BSW animals. A variant filtration approach that takes into account potential sequencing errors, flawed genotypes in animals with low sequencing coverage, and misclassified haplotypes (see Material and Methods), yielded 824 variants that were compatible with the inheritance pattern of the top haplotype (S3 Table). Of the compatible variants, 126 were located within the 2.38 Mb segment of extended autozygosity.

The same filtration approach was applied to 57 larger sequence variants (insertions, deletions, duplications, inversions, translocations) that were detected between 55 and 65 Mb on BTA6 in the 42 BSW animals. However, none of these variants was compatible with the recessive inheritance of the top haplotype.

Because homozygosity for the top haplotype was associated with reduced semen quality and fertility, we hypothesized that the causal variant might reside in the coding sequence of a gene that is expressed in the testis tissue. Only four of 824 compatible variants were located in protein-coding regions: three synonymous variants in *TLR6* encoding Toll-like receptor 6 (rs68268274 at Chr6:58069459), in *WDR19* encoding WD repeat-containing protein 19 (rs474302732 at Chr6:58373887) and in *GABRA2* encoding Gamma-aminobutyric acid receptor subunit alpha-2 (rs42595907 at Chr6:64905616), and a missense variant in *FAM114A1* encoding family with sequence similarity 114 member A1 that had a SIFT score of 0.18 indicating that this amino acid change is tolerated (rs382246003 at Chr6:58099868, ENSBTAP00000044312.1:p.Asn22Lys) (S3 Table). The *TLR6*, *WDR19* and *FAM114A1* genes are within the 2.38 Mb segment of extended autozygosity. *TLR6*, *WDR19* and *FAM114A1* are expressed at 4, 16 and 27 TPM, respectively, in testicular tissue of mature bulls (S2 Fig). The *GABRA2* gene is not expressed (TPM < 1) in testicular tissue of mature bulls. Moreover, the synonymous variant in *GABRA2* was more than 5 Mb from the segment of extended autozygosity, suggesting that it is less likely causal for the reduced semen quality and fertility of homozygous BSW bulls. We did not detect any sequence variants nearby the *NSUN7* gene neither in coding nor in non-coding sequences that were compatible with the inheritance pattern of the top haplotype (Fig 3C and S3 Table).

Using data from our in-house variant database and the latest variant discovery and genotyping run (run 7) of the 1000 Bull Genomes Project (http://www.1000bullgenomes.com/), we investigated the genotype distribution of the compatible variants in cattle from various breeds (S3 Table and S4 Table). It turned out that the variants in *TLR6*, *GABRA2* and *FAM114A1* frequently occur in either heterozygous or homozygous state in cattle from various breeds including Fleckvieh. Such variants are not likely to be causal, because no QTL for semen quality and bull fertility had been detected in Fleckvieh nearby the top haplotype on bovine chromosome 6 [25,28]. In contrast, the Chr6:58373887 T-allele that is located in the coding sequence of *WDR19* was only detected in BSW cattle and in the heterozygous state in one bull (SAMEA5064547) of the Nordic Red Dairy cattle breed. The Nordic Red Dairy cattle breed has recently experienced considerable introgression of BSW haplotypes [38].

## The BTA6:58373887 T-allele in *WDR19* activates cryptic splicing

A closer inspection of the BTA6:58373887C>T variant in the *WDR19* gene (ENSBTAG00000014512) revealed that it is located at the 3' end of exon 12, 8 bp from the splice donor site of intron 12. Although annotated as synonymous variant, we hypothesized that the T-allele might activate a novel exonic splice donor site (Fig 4A and 4B). *In silico*

prediction corroborated that the BTA6:58373887 T-allele very likely activates cryptic splicing (prediction score: 0.88) that eliminates 9 bp from exon 12 and causes the in-frame deletion of 3 amino acids in the WD repeat-containing domain of WDR19. WDR19 is part of the intraflagellar transport complex A that is essential for the physiological function of motile cilia and flagella [39,40]. Thus, we considered the BTA6:58373887 T-allele in the *WDR19* gene as a plausible candidate mutation for the reduced semen quality and fertility of the BSW bulls.

To investigate functional consequences of the BTA6:58373887C>T variant, we sampled testicular tissue of one wild-type, one heterozygous and two homozygous mutant AI bulls from which between 24 and 93 ejaculates had been collected and analysed. The average proportion of motile sperm was 40 and 69% in ejaculates of the two homozygous bulls. Sperm motility was higher in non-carrier (80%) and heterozygous (84%) bulls. The proportion of ejaculates that contained many sperm with either head or tail anomalies (scores 2 and 3) was clearly higher in the two homozygous bulls (40 and 88%) than the heterozygous (0%) or non-carrier bull (2%).

Histological sections of testicular tissue of two homozygous bulls showed no apparent pathological structures that might cause reduced semen quality. We extracted RNA from testicular tissue in order to examine *WDR19* transcription by reverse transcription PCR (RT-PCR). Using primers located in exons 12 and 13, we obtained a 169 bp RT-PCR product in the wild-type bull (Fig 4A, 4B and 4C). In bulls homozygous for the mutant BTA6:58373887 T-allele, we primarily detected 160 bp RT-PCR products. Sequence analysis of the 160 bp RT-PCR products confirmed that the mutant T-allele activates a cryptic exonic splice donor site resulting in 9 bp shorter sequence of exon 12 (Fig 4C). We also detected a weak band at 169 bp in homozygous animals indicating the presence of the wild-type length RT-PCR product. Sequence analysis confirmed that bulls homozygous for the BTA6:58373887 T-allele express the wild-type length transcript at low levels. Both the wild-type and mutant RT-PCR products were detected at approximately similar amounts in the heterozygous bull.

In order to quantify the abundance of the two WDR19 isoforms, we generated 63 million 150 bp sequencing reads from RNA prepared from testis tissue sampled from one BSW bull (sample accession number SAMN14485268) homozygous for the BTA6:58373887 T-allele. Of 29 RNA-seq reads that overlap the junction of exons 12 and 13, 5 and 24 correspond to the wild-type and mutant transcript, respectively, indicating that the wild-type and mutant isoforms are expressed at a ratio of 1:5 in the bull homozygous for the BTA6:58373887 T-allele (S3 Fig).

The mRNA expression pattern of *NSUN7* was similar in the BSW bull homozygous for the BTA6:58373887 T-allele and two Angus bulls homozygous for the wild-type C-allele (S4 Fig).

Bioinformatic analysis of the sequences of the RT PCR products revealed that the altered sequence of exon 12 leads to the in-frame deletion of three evolutionarily conserved amino acids (positions 414–416) of the WDR19 protein (ENSBTAT00000019294.6) (Fig 4F). The affected amino acids are located in the tenth WD repeat unit. The elimination of three amino acids is predicted to alter hotspot residues of WDR19 that are supposed to be relevant for protein-protein interactions and the assembly of functional complexes [41] (Fig 4D).

Next, we analysed the effect of the alternative exon 12 sequence on the *WDR19* mRNA and protein expression in testicular tissue. Quantitative PCR analysis with primers located in exons 15 and 16 revealed no differences among the analysed samples indicating that the BTA6:58373887 T-allele does not affect the *WDR19* mRNA expression. However, western blot analysis revealed that the amount of WDR19 is reduced in testicular tissue of heterozygous and even stronger in homozygous mutant bulls, indicating that the BTA6:58373887 T-allele is associated with WDR19 protein production (Fig 4E).

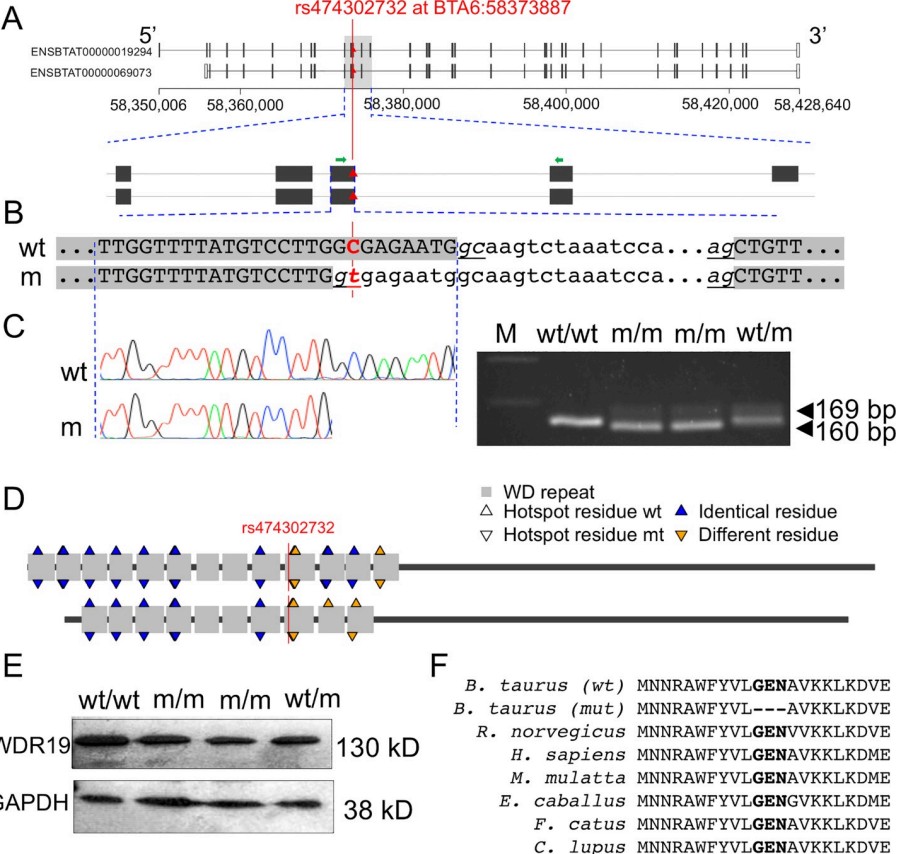

**Fig 4. Effect of the BTA6:58373887 C/T variant on WDR19 mRNA and protein expression.** (A) Two WDR19 isoforms with 1342 (ENSBTAT00000019294.6) and 1242 (ENSBTAT00000069073.1) amino acids are annotated in cattle. The BTA6:58373887C>T variant is located at the 3' end of exon 12 (ENSBTAT00000019294.6), 8 basepairs from the splice donor site of intron 12. The red triangle indicates the 58373887C>T variant, and green arrows indicate RT-PCR primers. (B) Genomic sequence surrounding the 58373887C>T-variant. Grey background indicates exons of the wild type (wt) and mutant (mt) transcripts, respectively. Underlined nucleotides indicate splice donor and splice acceptor sites. Upper and lower case letters indicate exonic and intronic nucleotides, respectively. (C) RT-PCR analysis on testis tissue samples from wild type, mutant and heterozygous bulls. Resequencing of the transcripts confirmed that the BTA6:58373887 T-allele activates a novel exonic splice site that is predicted to eliminate 3 amino acids from the resulting protein. (D) The mutation is located in the WD repeat domain of WDR19. The long (ENSBTAT00000019294.6, upper figure) and short (ENSBTAT00000069073.1, lower figure) WDR19 isoform contains 13 and 10 WD repeats, respectively. Residues at predicted amino acid hotspot positions differ between the wild-type and mutated protein for both WDR19 isoforms. (E) Western Blot analysis in wild type, mutant and heterozygous bulls. GAPDH was used as the control. (F) Multi-species alignment of the WDR19 protein sequence. Bold type indicates residues that are missing in the mutated (mt) WDR19 protein. Protein sequences were obtained from Ensembl for *Bos taurus* (ENSBTAT00000019294.6), *Rattus norvegicus* (ENSRNOT00000003991.6), *Homo sapiens* (ENST00000399820.8), *Macaca mulatta* (ENSMMUT00000003922.4), *Equus caballus* (ENSECAT00000026479.2), *Felis catus* (ENSFCAT00000060823.2) and *Canis lupus familiaris* (ENSCAFT00000025574.4).

## The QTL on BTA6 can be validated in BSW bulls from Germany and Austria

In order to validate the effect of the fertility-associated haplotype in an independent population, we analysed genotype and fertility data of 2481 BSW bulls from Germany and Austria. The German, Austrian and Swiss BSW bulls are genetically connected. However, the bulls from Austria and Germany were housed at different AI centers than the Swiss bulls and their fertility was estimated using a separate genetic evaluation system, thus they qualify as validation cohort. Ejaculate data were not available for the German and Austrian bulls precluding the analysis of semen quality.

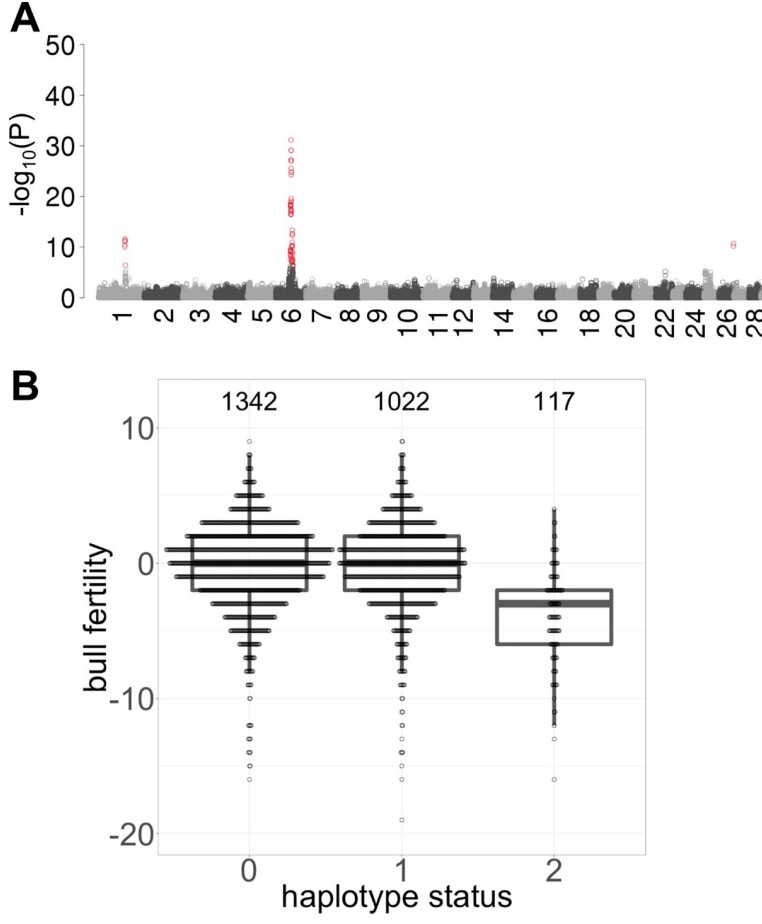

**Fig 5. Detection of QTL for male fertility in 2481 Austrian and German BSW bulls.** (A) Manhattan plot representing the association (-$\log_{10}$(P)) of 112,667 haplotypes with male fertility assuming a recessive mode of inheritance. Red color indicates significantly associated haplotypes (P < Bonferroni-corrected significance threshold). (B) The fertility of homozygous haplotype carriers is reduced by 1.01 phenotypic standard deviations.

The haplotype-based association study (recessive model) yielded strong association of the QTL on BTA6 (Fig 5A). An inflation factor of 1.03 indicated that population stratification was properly taken into account. On BTA6, we detected 69 haplotypes between 55,423,341 and 64,927,963 bp that exceeded the Bonferroni-corrected significance threshold (P < 4.4 x $10^{-7}$). The most significantly (P = 7.09x$10^{-32}$) associated haplotype was located between 57,335,668 and 57,624,763 bp. The frequency of this haplotype was 26% in the German and Austrian BSW population, i.e. slightly higher than the frequency of the top haplotype in the Swiss population. 127 and 1022 bulls from Germany and Austria were homozygous and heterozygous haplotype carriers, respectively.

The haplotype that showed the strongest association in the Swiss population (between 57,538,068 and 57,993,128 bp) was the second top haplotype in the German and Austrian BSW population and the P value was only slightly higher (P = 7.8x$10^{-30}$) compared to the top haplotype. The fertility of homozygous bulls was lower than that of heterozygous and non-carrier bulls (Fig 5B). The association signal on chromosome 6 was absent when the association analysis was conditioned on the top haplotype from the Swiss population, indicating that the BTA6:58373887 T-allele also tags the QTL detected in the German and Austrian BSW population.

## Discussion

Our haplotype-based association study uncovered a QTL on BTA6 for semen quality and fertility in BSW cattle. Response variables for genome-wide association testing were derived from 33 ejaculates per bull in average. Thus, our results confirm that variation in semen quality is partly due to genetic differences that can be identified in genome-wide association studies [42]. It is well known that bulls suffering from undetected diseases and stress may temporarily produce ejaculates that do not fulfill the requirements for artificial insemination [43,44]. Bulls that produce ejaculates of low quality despite being healthy might carry rare genetic variants that can be identified using case-control association testing [29,30]. We considered only ejaculates that fulfill minimum requirements for AI. The exclusion of ejaculates with low semen quality likely removes phenotypic outliers that could lead to false-positive association signals [45]. Low genomic inflation factors indicated that the results of our association studies were not enriched for spurious associations. However, we detected only very few haplotypes on chromosomes other than BTA6 that exceeded the Bonferroni-corrected significance threshold. Our rigorous approach to retain only ejaculates that comply with the requirements of AI may have reduced the amount of genetic variation that is amenable to association testing, thus possibly reducing the statistical power to detect QTL with small effects. Moreover, the heritability was less than 0.26 for all semen quality traits considered, which corroborates previous results in cattle [9–12]. Repeatability was higher than heritability indicating that the semen quality of AI bulls is markedly influenced by permanent environmental effects that may also capture non-additive genetic effects [46], which agrees with previous findings in boars and bulls [9], [47]. Detecting QTL with small to medium effects requires large mapping populations particularly for traits with low heritability [31]. Our mapping cohort consisted of 794 BSW bulls that were used for AI in the past 20 years. The size of this cohort is too small to detect QTL for semen quality that are either rare or explain only a small fraction of the trait variation. Thus, our findings suggest that, apart from the QTL on BTA6, variants with small effects contribute to the genetic variance of semen quality from BSW bulls, otherwise they should have been detected in our study. The analysis of data from multiple countries might increase the statistical power to detect trait-associated variants and provide additional insight into the genetic determinants of quantitative variation in semen quality. However, this requires a uniform assessment of semen quality across AI centers.

The QTL detected on BTA6 acts in a recessive manner. Ejaculates of bulls homozygous for the top haplotype contain an increased proportion of sperm with head and tail anomalies and lower proportion of motile sperm. However, the semen quality of homozygous bulls is only slightly reduced in comparison to the average semen quality of BSW bulls. Thus, the BTA6 QTL contributes to the quantitative variation in semen quality and male fertility. Most ejaculates of bulls that are homozygous for the top haplotype fulfill the requirements for AI. Because the reduced semen quality of homozygous bulls is recognized during the routine quality assessment at the AI center, the insemination straws produced from their ejaculates contained more sperm per dose than standard doses. Increasing the number of sperm per dose typically increases the fertilization rates for ejaculates that contain sperm with compensable defects [15]. However, the insemination success of homozygous bulls is significantly reduced, indicating that an increased number of sperm per dose is not sufficient to enable normal fertility in homozygous haplotype carriers. Moreover, ejaculates from homozygous bulls are more often discarded due to insufficient semen quality than ejaculates produced by either heterozygous or non-carrier bulls. Because we did not consider discarded ejaculates to calculate average semen quality, the actual effect of the haplotype on semen quality might be greater than estimated in our study. Nevertheless, it is important to emphasize that homozygous bulls are fertile. Therefore, the consequences

of the BTA6 haplotype on male reproduction are less detrimental than, e.g., of recessive haplotypes that severely compromise semen quality and lead to male infertility [28–30].

We make the remarkable observation that the BTA6 haplotype compromising male fertility segregates in the BSW population at a frequency of 24%. Balancing selection may maintain deleterious alleles at high frequency in natural populations [48]. However, if the haplotype has desirable effects on economically relevant traits, the effects must be very small; otherwise, they would have been detected in a recent study that carried out genome-wide association studies for more than 50 economically important traits in 4578 BSW bulls [49]. We can not exclude the possibility that the haplotype affects traits that are not routinely recorded in BSW cattle. Hitchhiking with favourable alleles may also explain the high frequency of the deleterious haplotype on BTA6 [50]. Yet, the haplotype is more than 20 million basepairs from the *NCAPG* gene and the casein cluster, which are targets of selection for stature and milk production in many dairy cattle breeds [51,52]. Signatures of selection were not detected in BSW cattle nearby the male fertility-associated QTL on BTA6 [49,53]. Random drift and the frequent use of haplotype carriers in AI most likely propagated the haplotype in spite of its negative effect on male reproduction. Founder effects caused the frequent phenotypic manifestation of deleterious recessive alleles also in the Fleckvieh and Belgian Blue cattle breeds [28,54]. Another reason that likely contributed to the high haplotype frequency is the fact that reduced fertility becomes evident only in homozygous AI bulls. Nevertheless, close monitoring of the haplotype seems warranted in order to prevent a further increase in frequency and the emergence of many homozygous bulls with low semen quality and fertility.

Low effective population size and the resulting long range linkage disequilibrium typically result in large QTL confidence intervals that contain many candidate causal variants. We detected significantly associated haplotypes that were in linkage disequilibrium with the top haplotype within a 15 Mb interval on chromosome 6 (Fig 3A). Multi-breed association analysis can be used to refine QTL because linkage disequilibrium is conserved over shorter distances across breeds than within breeds. However, association testing of semen quality and fertility across breeds was not possible because ejaculate data were only available for BSW bulls. Using whole-genome sequencing data of 42 BSW cattle with known haplotype status at the BTA6 QTL, we identified 824 variants that were compatible with the inheritance of the top haplotype. Variants segregating in breeds other than BSW can be discarded as candidate causal variants if the trait-associated mutation occurred after the formation of breeds. A filtration approach based on this assumption has been frequently applied to detect causal variants for recessive conditions in livestock and companion animals [37,55,56]. The removal of variants that segregate in breeds other than BSW would have excluded many of the 824 variants as potential candidate causal variants for the impaired fertility associated with homozygosity for the BTA6 QTL. Nevertheless, such a variant filtration approach does not take into account that deleterious alleles may segregate across populations [57–60]. It is possible that the QTL detected in our study also segregates in breeds other than BSW. However, QTL for semen quality and bull fertility were not detected in Fleckvieh cattle nearby the BTA6 QTL [25,28]. Thus, variants that occur at high frequency also in the Fleckvieh population are less likely to be causal for impaired reproductive performance. We applied an appropriate variant filtration approach to prioritize a variant (BTA6:58373887) in the coding sequence of the *WDR19* gene. Although annotated as synonymous variant, our analyses demonstrated that the BTA6:58373887 T-allele activates a novel exonic splice site that eliminates 3 amino acids from the resulting protein. Thus, our results indicate that nondescript synonymous variants warrant close scrutiny because they might affect exonic splicing enhancers [61] or activate cryptic splice sites, which is not immediately apparent from standard variant annotation as it was the case for the BTA6:58373887 T-allele in *WDR19*.

The WD repeat-containing protein 19 is part of the intraflagellar transport complex A (IFT-A) that is essential for the assembly and physiological function of motile cilia and flagella [39,40,62,63]. Loss-of-function mutations in the WD repeat domain of the *WDR19* gene lead to severe ciliopathies in humans [64–67]. The consequences of these mutations on semen quality and male fertility are unknown, because affected individuals do not reach reproductive age [68]. The BTA6:58373887 T-allele is not a loss-of-function mutation because we detected the allele in the homozygous state in mature bulls, which were healthy apart from producing ejaculates with reduced semen quality and fertility. However, an excess of sperm with morphological anomalies and low motility indicates that spermatogenesis of bulls homozygous for the BTA6:58373887 T-allele is impaired. Our results show that *WDR19* is expressed in the testes of mature bulls. The BTA6:58373887 T-allele activates alternative splicing that eliminates 3 amino acids from the resulting protein. The loss of evolutionary conserved amino acids within the WD repeat domain alters hotspot residues at the surface of the WDR19 protein, which might compromise its secondary structure and interaction with other proteins including the constituents of the IFT-A complex [41,69]. We observed markedly reduced WDR19 protein expression in testicular tissue of heterozygous and homozygous bulls indicating that changes in the secondary structure may compromise protein stability and result in either protein degradation or post-translational modification of this WDR19 isoform. A lower amount of WDR19 may compromise spermiogenesis due to impaired intraflagellar transport thus causing abnormal sperm with low fertility in homozygous bulls. We also detected the wild-type length transcript at low levels in bulls homozygous for the BTA6:58373887 T-allele indicating differential splicing. Factors controlling splice site selection may contribute to the marked phenotypic variability in the semen quality of bulls homozygous for the BTA6:58373887 T-allele. The presence of a low amount of wild-type WDR19 protein seems to be sufficient to produce normal sperm. However, semen quality of individuals that exceed a certain threshold of mutant protein or undercut a certain threshold of wild-type protein might have a higher proportion of abnormal sperm, thus resulting in lower fertility.

This is the first report providing evidence that naturally occurring variation in *WDR19* is associated with reduced semen quality and fertility. However, variants affecting other constituents of the IFT-A complex have been shown to compromise male reproduction. Mice with a loss-of-function variant in the *IFT140* gene show multiple morphological aberrations of the sperm including short and/or bent sperm tail and abnormal heads [70]. Compound heterozygosity in human *IFT140* is associated with reduced sperm count and an excess of sperm with head and tail anomalies in an otherwise healthy individual [71]. This indicates that mutations of the IFT-A complex may reduce semen quality without compromising the general health, which agrees with our findings of a deleterious allele in bovine *WDR19* that affects semen quality and fertility in otherwise healthy bulls.

## Methods

### Ethics approval and consent to participate

Semen was collected at approved AI centers. Testis tissue was collected after regular slaughter at an approved slaughterhouse. The decision to slaughter the bulls was independent from our study. None of the authors of the present study was involved in the decision to slaughter the bulls. No ethics approval was required for the analyses.

### Consent for publication

Breeding associations and AI centers provided written consent to the analyses performed and the publication of results and data.

## Genotypes of Brown Swiss bulls

Genotypes of 4549 BSW bulls were provided by Swiss, German and Austrian breeding associations. A subset (N = 870) of the bulls was genotyped using the Illumina BovineHD (HD) bead chip that comprises 777,962 SNPs. All other bulls (N = 3679) were genotyped at approximately 50,000 SNPs using medium density (MD) chips (e.g., the Illumina BovineSNP50 bead chip that comprises 54,001 (version 1) or 54,609 (version 2) SNPs). The position of the SNPs corresponded to the ARS-UCD1.2-assembly of the bovine genome [72]. Quality control on the genotype data was carried out separately for the HD and MD datasets using the *plink* (version 1.9) software [73]. Animals and SNPs with more than 20% missing genotypes were excluded from the data. We removed SNPs with minor allele frequency (MAF) less than 0.005 and SNPs for which the observed genotype distribution deviated significantly (P < 0.00001) from Hardy-Weinberg proportions. After quality control, 36,131 and 607,511 SNPs remained in the MD and HD dataset, respectively. Sporadically missing genotypes were imputed in the MD and HD datasets separately using the *Beagle* (version 5.0) software [74]. Subsequently, we inferred haplotypes for both datasets using the *Eagle* (version 2.4) software [75]. We considered the HD genotypes of 870 bulls as a reference to impute MD genotypes to higher density using the haplotype-based imputation approach implemented in the *Minimac3* (version 2.0.1) software [76]. Following imputation, the genotype panels were merged to obtain the final dataset consisting of 4549 bulls genotyped at 607,511 autosomal SNPs. Principal components of the genomic relationship matrix were calculated using the *plink* (version 1.9) software [73].

## Semen quality data of Swiss BSW bulls

The Swiss AI center Swissgenetics provided data on 70,990 ejaculates that were collected between January 2000 and March 2018 from 1,343 BSW bulls at the AI center in Mülligen, canton of Aargau, Switzerland. All ejaculates were collected as part of the breeding and reproduction service of the AI center to Swiss cattle farming. Semen quality was assessed by lab technicians immediately after ejaculate collection in order to identify and discard ejaculates of low quality. The parameters recorded for fresh ejaculates were semen volume (in ml), sperm concentration (million sperm cells per ml) quantified using photometric analysis, and sperm motility (percentage of sperm with forward motility) assessed visually using a heated-stage microscope at 200-fold magnification. Moreover, the presence of erythrocytes, leucocytes and other non-sperm cells was documented for each ejaculate. Each ejaculate received a score between 0 and 3 indicating the proportion of sperm with head and tail anomalies (0: no or very few anomalies, 1: less than 10% sperm with anomalies, 2: between 10 and 30% sperm with anomalies, 3: more than 30% sperm with anomalies). Ejaculates that fulfilled minimum requirements for artificial insemination (semen volume above 1 ml, more than 300 million sperm per ml, at least 70% motile sperm, no apparent impurities and no excessive abnormalities of sperm heads and tails) were diluted using a Tris-egg yolk based extender, filled in straws containing between 15 and 25 million sperm cells and cryoconserved in liquid nitrogen.

We removed records of ejaculates that were pooled before semen analysis and considered only the first ejaculate per day that was collected from bulls between 400 and 1000 days of age (sometimes more than one ejaculate is collected per bull and day). Records for which either the interval between successive ejaculates or the semen collector were missing were excluded from our analysis. We retained only ejaculates that complied with minimum requirements for artificial insemination because insufficient semen quality might be attributable to rare genetic conditions [28,29]. Eventually, we retained only ejaculates of bulls for which at least 8 ejaculates were available. Our final dataset contained 26,090 ejaculates for 794 genotyped bulls (ø 33 ± 29 ejaculates per bull; median: 20) (Table 2). We considered six semen quality traits for

**Table 2. Quality control on the raw semen quality data.**

| Filtering parameter | Number of ejaculates | Number of BSW bulls |
|---|---|---|
| Raw dataset | 70,990 | 1343 |
| Interval to preceding ejaculate known (excluding first ejaculate of bull in dataset) | 69,647 | 1343 |
| Age at semen collection between 400 and 1000 days | 56,200 | 1302 |
| Ejaculate volume recorded | 47,956 | 1294 |
| Fresh sperm motility recorded | 47,941 | 1294 |
| No (biological / technical) cause of rejection recorded | 43,733 | 1250 |
| Fresh sperm motility $\geq$ 70% | 43,571 | 1235 |
| Ejaculate was not pooled before semen analysis | 35,888 | 1224 |
| Only first ejaculate per day | 32,584 | 1221 |
| Semen collector recorded | 30,966 | 1201 |
| Plausible sperm head and tail anomaly score | 30,964 | 1201 |
| Target number of sperm in insemination straw between 15 and 25 million | 30,832 | 1201 |
| At least 8 ejaculates collected per bull | 29,751 | 946 |
| Bull had (partially imputed) genotypes | 26,090 | 794 |

our genetic investigations: ejaculate volume, sperm concentration, sperm motility, sperm head and tail anomaly score, and number of sperm filled per insemination straw.

## Estimation of genetic parameters for semen quality

The heritability and repeatability of six semen quality traits was calculated using the average information restricted maximum likelihood (REML) estimation algorithm implemented in the *AIREMLF90* program [77]. We fitted the following univariate linear mixed model: $\mathbf{y} = \mathbf{1}\mu + \mathbf{h}a + \mathbf{d}f + \mathbf{C}t + \mathbf{S}n + \mathbf{Z}_u\mathbf{u} + \mathbf{Z}_p\mathbf{p} + \mathbf{e}$, where $\mathbf{y}$ is a vector of each phenotype tested, $\mu$ is the intercept ($\mathbf{1}$ is a vector of ones), $\mathbf{h}$ and $\mathbf{d}$ are vectors of age and interval between successive semen collections in days and a and f are their respective effects; $\mathbf{C}$, $\mathbf{S}$, $\mathbf{Z}_u$ and $\mathbf{Z}_p$ are incidence matrices relating collector (person collecting the ejaculate), season (4 seasons per year), random individual polygenic and random permanent environment effects to their respective records respectively, $\mathbf{t}$ is the vector of effects of different semen collectors, $\mathbf{n}$ is the vector of effects of different seasons, $\mathbf{u}$ is the vector of random individual polygenic effects assumed to be normally distributed ($N(0, \mathbf{A}\sigma_g^2)$, where $\mathbf{A}$ is the additive relationship matrix estimated from the pedigree and $\sigma_g^2$ is the additive genetic variance), $\mathbf{p}$ is the vector of permanent environmental effects assumed to be normally distributed ($N(0, \mathbf{I}\sigma_{pe}^2)$, where $\sigma_{pe}^2$ is the permanent environment variance), and $\mathbf{e}$ is the vector of individual error terms assumed to be normally distributed ($N(0, \mathbf{I}\sigma_e^2)$, where $\sigma_e^2$ is the residual variance). Standard errors of heritability and repeatability were estimated using the Monte-Carlo method [78] implemented in *AIREMLF90*.

## Semen quality phenotypes for association testing

Phenotypes for the association studies were the average values either from the filtered unadjusted data or from the residuals of the following linear model: $\mathbf{y} = \mathbf{1}\mu + \mathbf{h}a + \mathbf{d}f + \mathbf{C}t + \mathbf{S}n + \mathbf{e}$, where $\mathbf{y}$ is a semen quality parameter of each ejaculate, $\mu$ is the intercept, 1 is a vector of ones, $\mathbf{h}$ and $\mathbf{d}$ are vectors of the bulls' age (in days when the ejaculate was collected and the interval (in days) to the preceding ejaculate, respectively, a and f are their respective effects. $\mathbf{C}$ and $\mathbf{S}$ are

incidence matrices relating the person collecting the ejaculate and semen collection date (4 seasons per year), t and n are their respective effects and **e** is a vector of random residuals. The correlation between the average values from the filtered data and the average values from the random residuals were high (r > 0.90) for all traits analysed. To facilitate a better interpretation of the effect estimates, we used the average values calculated from the filtered data as input variables throughout the manuscript.

## Insemination success in the Swiss BSW population

Bull fertility (as at February 2019) was provided by Swissgenetics for 941 BSW bulls that had between 232 and 15,690 first inseminations with conventional (i.e., semen was not sorted for sex) frozen-thawed semen. Male fertility was estimated using a linear mixed model similar to the one proposed by Schaeffer et al. [79]: $\mathbf{y}_{ijklmnopq} = \mu + \mathbf{MO}_i + \mathbf{PA}_j + \mathbf{PR}_k + (\mathbf{RS} \times \mathbf{RK})_{lm} + \mathbf{BE}_n + h_o + \mathbf{s}_{lp} + \mathbf{e}_{ijklmnopq}$, where $\mathbf{y}_{ijklmnopq}$ is either 0 (subsequent insemination recorded within 56 days of the insemination) or 1 (no subsequent insemination recorded within 56 days after the insemination), $\mu$ is the intercept, $\mathbf{MO}_i$ is the insemination month, $\mathbf{PA}_j$ is the parity (heifer or cow), $\mathbf{PR}_k$ is the cost of the insemination straw, $(\mathbf{RS} \times \mathbf{RK})_{lm}$ is the combination bull's breed x cow's breed, $\mathbf{BE}_n$ is the insemination technician, $h_o$ is the herd, $\mathbf{s}_{lp}$ is the fertility of the bull expressed in % deviation from the average non-return rate, and $\mathbf{e}_{ijklmnopq}$ is a random residual term. Bull fertility was subsequently standardized to a mean of 100 ± 12. Three bulls with very low fertility (i.e., more than 3 standard deviations below average) were not considered for our analyses because they might carry rare genetic conditions [28] and including such phenotypic outliers in genome-wide association testing might lead to spurious associations [45]. For the GWAS on bull fertility, we considered 591 bulls that had records for both male fertility and semen quality.

## Male fertility in the German and Austrian BSW populations

Phenotypes for bull fertility (as at December 2017), were provided by ZuchtData EDV-Dienstleistungen GmbH, Austria, for 4617 BSW bulls from Germany and Austria that were used for 4,267,990 and 1,646,254 inseminations in cows and heifers, respectively. Bull fertility in the German and Austrian BSW populations is estimated using a multi-trait animal model that was proposed by Fuerst & Gredler [4]. The model includes a fixed effect for the service sire that represents bull fertility as deviation from the population mean. Four phenotypic outliers were not considered for subsequent analyses (see above). The final dataset included fertility records for 2481 bulls that also had (partially imputed) genotypes at 607,511 SNPs.

## Haplotype-based association testing of phenotypes

Haplotype-based association testing was implemented in R using a sliding-window-based approach that we applied previously to investigate genetic conditions in cattle [58]. In brief, a sliding window of 50 contiguous SNPs corresponding to a haplotype length of ~200 kb was shifted along the autosomes in steps of 15 SNPs. Within each sliding window ($N$ = 40,444), pre-phased haplotypes (see above) with frequency greater than 1% were tested for association using the linear model $\mathbf{y} = \mu + \sum_{j=1}^{10} \mathbf{a}_j \mathbf{PC}_j + \mathbf{bHT} + \mathbf{e}$, where y is a vector of phenotypes (see above), $\mu$ is the intercept, $\mathbf{PC}_j$ are the top ten principal components of the genomic relationship matrix (see above), **a** and **b** are effects of the principal components and the haplotype (**HT**) tested, respectively, and **e** is a vector of residuals that are assumed to be normally distributed. Haplotypes were tested for association assuming either additive, dominant or recessive mode of inheritance. Recessive tests were carried out for haplotypes that were observed in the

homozygous state in at least 1% of the individuals. The genomic inflation factor lambda was calculated in R using the following formula: lambda = median(qchisq(1-p, 1)) / qchisq(0.5, 1) where p is a vector of P values.

## Whole-genome sequencing and sequence variant genotyping

We used paired-end whole-genome sequencing reads (2x101 bp, 2x126 bp or 2x150 bp) of 42 BSW animals (3 homozygous, 10 heterozygous, 29 non-carrier) that were generated using Illumina HiSeq or NovaSeq instruments to identify candidate causal variants for the impaired reproductive performance of homozygous haplotype carriers. Some sequenced animals were key ancestors of the Swiss BSW population that had been sequenced previously [80]. Sequencing data of all animals are available from the European Nucleotide Archive (http://www.ebi.ac. uk/ena) under the accession numbers listed in S4 Table.

We removed from the raw data reads for which the phred-scaled quality was less than 15 for more than 15% of bases, and trimmed adapter sequences using the *fastp* software [81]. Subsequently, the sequencing data were aligned to the bovine linear reference genome (ARS-UCD1.2) using the *BWA mem* algorithm [82]. Duplicates were marked and read alignments were coordinate sorted using the *Picard* tools software suite [83] and *Sambamba* [84], respectively. The average depth of the aligned sequencing reads was 13.9-fold and it ranged from 6.6 to 28.9-fold.

We discovered and genotyped SNPs and Indels from the linear alignments using the multisample variant calling approach implemented in the *Genome Analysis Toolkit* (*GATK*, version 4.1.0) [85]. Specifically, we followed *GATK*'s best practice recommendations for sequence variant discovery and filtration when variant quality score recalibration (VQSR) is not possible because the truth set of variants required for VQSR is not publicly available in cattle. Additional details on the applied variant calling and filtration approach can be found in Crysnanto et al. [86]. The *mosdepth* software (version 0.2.2) [36] was used to extract the number of reads that covered a genomic position.

Structural variants including large insertions, deletions, inversions, duplications and translocations were detected and genotyped in the sequenced BSW animals using the *delly* software with default parameter settings [87].

## Identification of candidate causal variants

The status of the BTA6 top haplotype was determined for the 42 sequenced BSW animals using (partially) imputed SNP microarray-derived genotypes at 607,511 SNPs (see above). Sequence variant genotypes within a 10 Mb region on bovine chromosome 6 (from 55 to 65 Mb) encompassing the 2.38 Mb segment of extended homozygosity were filtered to identify variants compatible with recessive inheritance. In order to take into account possible haplotype phasing errors, inaccurately genotyped sequence variants and the undercalling of heterozygous genotypes due to low sequencing coverage, we applied a conservative filtering strategy. Specifically, we screened for variants that had following allele frequencies:

- $\geq 0.8$ in three homozygous haplotype carriers (5 out of 6 alleles),

- between 0.4 and 0.6 in ten heterozygous haplotype carriers,

- $\leq 0.05$ in 29 non-carriers of the haplotype.

This filtration identified 824 variants that were compatible with recessive inheritance. We annotated the candidate causal variants according to the Ensembl (release 95) annotation of the ARS-UCD1.2-assembly of the bovine genome using the *Variant Effect Predictor* (VEP)

software tool [88]. The frequency and genotype distribution of candidate causal variants were also investigated in cattle from breeds other than BSW using polymorphism information from an in-house sequence variant database of 295 cattle (accession numbers and corresponding breeds are listed in S4 Table) and the most recent release (run 7) of the 1000 Bull Genomes Project (3078 cattle).

## Quantification of transcript abundance in testis tissue

We downloaded between 47 and 58 million 2x150 bp paired-end sequencing reads from the ENA sequence read archive that were generated using RNA extracted from testicular tissue samples of three mature bulls and three newborn male calves of the Angus beef cattle breed (ENA accession numbers: SAMN09205186-SAMN09205191; [89]). The RNA sequencing data were pseudo-aligned to an index of the bovine transcriptome (ftp://ftp.ensembl.org/pub/release-98/fasta/bos_taurus/cdna/Bos_taurus.ARS-UCD1.2.cdna.all.fa.gz) and transcript abundance was quantified using the *kallisto* software [90]. We used the R package tximport [91] to aggregate transcript abundances to the gene level.

## Testicular tissue sampling

Testicular tissue was sampled at a commercial slaughterhouse from four AI bulls (two bulls homozygous for the mutant (mt) T-allele at Chr6:58373887, one heterozygous bull and one bull homozygous for the wild-type (wt) C-allele Chr6:58373887). At the time of slaughter, the bulls were between 595 and 778 days old and they had been kept under identical conditions at the AI center Swissgenetics. The haplotype status of the bulls was determined before slaughter using SNP microarray-derived genotypes. Tissue samples were frozen on dry ice immediately after collection and subsequently stored at –80˚C. Additionally, testis and epididymis tissues were formalin-fixed and paraffin-embedded. Sections of embedded tissue were stained with hematoxylin and eosin for microscopic examination.

## Transcription analysis using RT-PCR

Total RNA from testicular tissue samples of the four bulls was extracted using Direct-zol RNA Mini Prep Kit (Zymo Research) according to the manufacturer's instructions. The integrity and concentration of RNA was analysed by agarose gel electrophoresis and using the Nano-drop ND-2000 spectrophotometer (Thermo Scientific), respectively. Total RNA was reverse transcribed using FastGene ScriptaseII (Fast Gene). RT-PCR was done with the GoTaq Poly-merase (Promega) using forward primer (5'-ACGTGGAGCCCAACTTTGTA-3') and reverse primer (5'-AGTGCAGACGCATAGTCAGAA-3'). The RT-PCR products were separated on a 3% agarose gel and the length of the products was analysed using Quantum (Vilber Lourmat). The sequence of the RT-PCR products was obtained using Sanger sequencing.

## Quantitative RT-PCR

200 ng total RNA was used to synthesise complementary DNA (cDNA) using FastGene Scriptase II (Fast Gene). Two-step qPCR experiments were performed using Fast SybrGreen Master-Mix (Applied Biosystems) using forward primer (5'-TGATTATCGACATCCCGTCA-3') and reverse primer (5'-GTCTGGAATCTCATAGGTAG-3') and run on an ABI 7500 thermocycler (Applied Biosystems). Primer specificity and capture temperature were determined by melt curve analysis. The relative expression difference between the genotypes in all tissues was calculated for each sample (ΔΔCT). All cDNA samples were assayed in triplicate and relative expression levels normalised to the *GAPDH* reference gene.

## Whole transcriptome sequencing and read alignment

A paired-end RNA library (2x150 bp) was prepared from total RNA from a testicular tissue sample of one BSW bull homozygous for the mutant T-allele at Chr6:58373887 using the Illumina TruSeq RNA Sample Preparation Kit (Illumina, San Diego, CA, USA). The library was sequenced using an Illumina NovaSeq6000 instrument. Quality control on 63,473,978 raw RNA sequencing reads was performed using the *fastp* software [81]. We removed adapter sequences and reads for which the phred-scaled quality was less than 15 for more than 15% of bases. The filtered reads (N = 63,138,436) were aligned to the ARS-UCD1.2 reference sequence and the Ensembl gene annotation (release 99) using the splice-aware read alignment tool *STAR* (version 2.7.3a) [92]. The *mosdepth* software (version 0.2.2) [36] was used to extract the number of RNA sequencing reads that covered a genomic position. The RNA sequencing data have been deposited at the Sequence Read Archive of the NCBI under sample and study accessions SAMN14485268 and PRJNA616249, respectively.

## Western blot

Total protein from testicular tissue was extracted using T-PER® Tissue Protein Extraction Reagent (Thermo Scientific) according to the manufacturer's instructions. Western blot was performed using iBind Western Blot System (Life Technologies). The bovine WDR19 protein was detected using rabbit abx316410 anti-human-WDR19 antibody (diluted 1:2000 in iBind solution) and horseradish peroxidase-labelled goat anti-rabbit sc-2005 (diluted 1:5000 in iBind solution). GADPH was detected using mouse monoclonal anti-GADPH #G8795, (diluted 1:3000) and rabbit anti-mouse IgG H&L (HRP) ab6728 (diluted 1:5000).

## Bioinformatic analyses of the mutant allele and WDR19 topology prediction

The *NNSPLICE* software tool (https://www.fruitfly.org/seq_tools/splice.html, [93]) was used to predict putative splice sites within 1000 bp on either side of the BTA6:58373887C>T-variant. Multi-species alignment of the WDR19 protein sequence was performed using *Clustal Omega* [94,95]. The topology of bovine WDR19 isoforms ENSBTAT00000019294.6 (1342 amino acids) and ENSBTAT00000069073.1 (1242 amino acids) was predicted using the *WD40-repeat protein Structure Predictor* version 2.0 (http://www.wdspdb.com/wdsp/ [96]).

## Supporting information

**S1 Table. Phenotypic correlations (off-diagonal) and heritability (diagonal) of the traits studied.** Correlation and heritability of seven traits relevant for semen quality and fertility. (DOCX)

**S2 Table. Accession numbers of 42 BSW animals.** The numbers listed indicate accession numbers from the sequence read archive of the European Nucleotide Archive (http://www.ebi.ac.uk/ena). The second and third column indicates the status of the sequenced animals with regard to the top haplotype and the sequencing coverage, respectively. (CSV)

**S3 Table. Variants compatible with recessive inheritance.** Functional consequences of 824 variants were predicted using the VEP software tool. The frequency of the alternate allele is presented for homozygous, heterozygous and non-carrier animals of the BSW cattle breed (n = 42), as well as for animals from breeds other than BSW (n = 253) that are part of our in-

house variant database.
(CSV)

**S4 Table. Accession numbers of 295 animals from different cattle breeds.** The numbers listed indicate accession numbers from the sequence read archive of the European Nucleotide Archive (http://www.ebi.ac.uk/ena). The second column indicates the breed of the sequenced animals (BSW–Brown Swiss; FV–Fleckvieh; HOL–Holstein; NRC–Nordic Red Dairy Cattle; OBV–Original Braunvieh; TGV–Tyrolean Grey).
(CSV)

**S1 Fig. Detection of QTL for semen quality and fertility in BSW bulls.** Manhattan plots representing the association ($-\log_{10}(P)$) of haplotypes with (A) ejaculate volume (genomic inflation factor lambda = 1.67), (B) sperm concentration (lambda = 1.29), (C) sperm motility (lambda = 1.35), (D) proportion of sperm with head anomalies (lambda = 1.10), (E) proportion of sperm with tail anomalies (lambda = 1.14), (F) sperm per straw (lambda = 1.22), and (G) bull fertility (lambda = 1.22) assuming an additive mode of inheritance. Red color indicates significantly associated haplotypes (P < Bonferroni corrected significance threshold).
(TIF)

**S2 Fig. Expression of genes located within the segment of extended homozygosity in testis tissue.** Transcripts per million (TPM) in testis tissue of three mature bulls (grey) and three newborn male calves (black). The horizontal line represents the median expression (5.9 TPM) of 22,372 genes. To improve readability, the expression for *RPL* is only shown in the inset.
(PNG)

**S3 Fig. Activation of cryptic splicing through the BTA6:58373887 T-allele.** Screen captures of IGV outputs from testis RNAseq alignments of a BSW bull (SAMN14485268) homozygous for the mutant (mt) T-allele (A) and two control bulls from the Angus breed (SAMN09205187, SAMN09205188) that are homozygous for the wild-type (wt) C-allele at Chr6:58373887 (B, C). The red bar indicates nine nucleotides that are truncated from exon 12 of *WDR19* in the BSW bull (A) due to cryptic splicing activated by the T-allele. A low number of sequence reads corresponding to the wild-type transcript were also detected in the bull homozygous for the BTA6:58373887 T-allele.
(PDF)

**S4 Fig. Expression of *NSUN7*.** Expression of *NSUN7* quantified using testis RNAseq alignments of a BSW bull (SAMN14485268) homozygous for the mutant (mt) T-allele (A) and two control bulls from the Angus breed (SAMN09205187, SAMN09205188) that are homozygous for the wild-type (wt) C-allele at Chr6:58373887 (B, C). The number of reads covering a genomic position was extracted from coordinate sorted BAM files using the *mosdepth* software and subsequently divided by the total number of reads (in million) mapped to transcripts.
(PDF)

**S1 Data. Data of 794 bulls used to detect the QTL on BTA6.** The archive contains phased genotypes (haplotypes_6) at 28,872 SNPs located on chromosome 6 (markers_6), the top 20 principal components (evecs) and the average sperm motility of 794 BSW bulls.
(ZIP)

**S2 Data. R script used to perform the haplotype-based association testing.** This archive contains the R script that was used to carry out the haplotype-based association testing, a README file that provides information how the script can be applied to analyse the raw data provided in S1 Data and a Jupyter notebook file (GWAS.html) that explains how to process

the output file.
(ZIP)

## Acknowledgments

We thank Braunvieh Schweiz for providing pedigree and genotype data of BSW bulls. We thank Swissgenetics for providing ejaculate data of BSW bulls. We thank Susanne Meese and Sarah Wyck for explaining semen collection and processing at Swissgenetics, and Sarah Wyck and Meenu Bhati for support in tissue sampling. We acknowledge the Arbeitsgemeinschaft Deutsches Braunvieh, Braunvieh Austria, Tierzuchtforschung Grub, Chair of Animal Breeding of TU München, the Institute of Animal Breeding from Bayerische Landesanstalt fuer Landwirtschaft and ZuchtData EDV Dienstleistungen GmbH for providing pedigree records, genotype and phenotype data for the Austrian and German BSW bulls. We thank the Functional Genomics Center Zurich for generating DNA and RNA sequencing data.

## Author Contributions

**Conceptualization:** Ulrich Witschi, Heinrich Bollwein, Hubert Pausch.

**Data curation:** Maya Hiltpold, Danang Crysnanto, Hubert Pausch.

**Formal analysis:** Maya Hiltpold, Guanglin Niu, Naveen Kumar Kadri, Danang Crysnanto, Mirjam Spengeler, Fritz Schmitz-Hsu, Christian Fuerst, Hermann Schwarzenbacher, Franz R. Seefried, Ruedi Fries, Hubert Pausch.

**Funding acquisition:** Heinrich Bollwein, Hubert Pausch.

**Investigation:** Maya Hiltpold, Guanglin Niu, Zih-Hua Fang, Frauke Seehusen, Ruedi Fries, Krzysztof Flisikowski.

**Methodology:** Naveen Kumar Kadri, Hubert Pausch.

**Resources:** Fritz Schmitz-Hsu, Ulrich Witschi, Angelika Schnieke, Krzysztof Flisikowski.

**Supervision:** Heinrich Bollwein, Krzysztof Flisikowski, Hubert Pausch.

**Visualization:** Maya Hiltpold, Guanglin Niu, Krzysztof Flisikowski, Hubert Pausch.

**Writing – original draft:** Maya Hiltpold, Hubert Pausch.

**Writing – review & editing:** Maya Hiltpold, Guanglin Niu, Naveen Kumar Kadri, Danang Crysnanto, Zih-Hua Fang, Mirjam Spengeler, Fritz Schmitz-Hsu, Christian Fuerst, Hermann Schwarzenbacher, Franz R. Seefried, Frauke Seehusen, Ulrich Witschi, Ruedi Fries, Krzysztof Flisikowski, Hubert Pausch.

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
