## [Decision Letter · Decision Letter 0]

17 Mar 2020

Dear Dr Pausch,

Thank you very much for submitting your Research Article entitled 'Activation of cryptic splicing in bovine WDR19 is associated with reduced semen quality and male fertility' to PLOS Genetics. Your manuscript was fully evaluated at the editorial level and by independent peer reviewers. The reviewers appreciated the attention to an important problem, but raised some substantial concerns about the current manuscript. Based on the reviews, we will not be able to accept this version of the manuscript, but we would be willing to review again a much-revised version. We cannot, of course, promise publication at that time.

If you decide to revise the manuscript for further consideration at PLOS Genetics, please aim to resubmit within the next 60 days, unless it will take extra time to address the concerns of the reviewers, in which case we would appreciate an expected resubmission date by email to plosgenetics@plos.org.

[LINK]

We are sorry that we cannot be more positive about your manuscript at this stage. Please do not hesitate to contact us if you have any concerns or questions.

Yours sincerely,

Moira K. O'Bryan

Associate Editor

PLOS Genetics

Hua Tang

Section Editor: Natural Variation

PLOS Genetics

Firstly apologies for the delay obtaining these reviews. This was due to difficulties in finding suitable reviewers. As indicated, both reviewers agree this manuscript is of a good quality with the potential to make an important contribution to the field. Please review this comments and comment on all. The majority are easily achieved. Reviewer 1 has suggested additional experiments which would certainly help to support the claims within the manuscript.

Reviewer's Responses to Questions

**Comments to the Authors:**

Reviewer #1: Based mostly on GWAS and impressive genomic analyses, the authors identified a synonymous SNP in WDR19 gene of Brown Swiss cattle. They present circumstantial evidence that the frequent minor allele of this SNP activates cryptic splicing resulting with a deleterious effect on semen quality and male fertility. If true, their work presents most-valuable evidence for the importance of this gene in male fertility. Besides their prepublication in Biorxiv, their report is the first one to demonstrate this importance and thus would be of interest to a wide-readership. Yet, as indicated in Major Concerns, another strong candidate gene for male fertility is within the haplotype sharing the 2.38 Mb segment of extended autozygosity. The authors should consider further experimentation to prove that expression of this other gene is intact in homozygotes for the deleterious haplotype.

Major Concerns:

1. The authors ignore substantial literature related to the Nsun7 gene, which resides in the critical segment that they identified. This gene that was initially identified in infertile male mice (Harris et al., 2007, Biol Reprod. 77:376-82) has a deletion mutation associated with sperm motility defect in infertile men (Khosronezhad et al., 2015, J Assist Reprod Genet. 32:807-15). Unlike WDR19, NSUN7 gene is highly expressed in the authors' RNA-seq meta-analysis of the testis.

2. Brown Swiss cattle are some what under represented in the Short Read Archive (SRA). Yet, six RNA-seq submissions from skin samples (ERX1545689-94) are available. Using 32 bp probes for the three possible sequences of relevant splice junction, I BLASTN searched these SRA submissions concluding that it is spliced as expected (179 hits found) with no indication of the frequent mutated allele. Thus, performing such RNA-seq experiment using a testis tissue sample from a homozygote for the haplotype associated with infertility would allow to simultaneously asses if NSUN7 expression is intact, while WDR19 is not. This evidence is much needed to support the authors' claim.

Minor Comments:

Line 79: "pathopysiology" better "pathophysiology".

Lines 101 & 202: " mimimum"/"miminum" should be "minimum".

Line 286: "WD repeat-contaning" should be "WD repeat-containing"

Line 472: "protein wich" should be "protein, which".

Line 662: "Strcutural" should be "Structural".

Reviewer #2: Hiltpold et al. have identified a cryptic exonic splice variant in the WDR19 gene in cattle and in homozygous condition resulted in lower semen quality and lower male fertility. They used a haplotype-based association mapping followed by searching the candidate variants using genome sequence data. The variant was validated in independent populations. Western blot analysis was done to study the protein expression, which further confirmed the functionality of this variant. The manuscript is well written and easy to follow. I have only a few minor comments.

Introduction part is too long and may be shortened.

The authors described the WDR19 variant had recessive gene effect. However, markedly reduced WDR19 protein expression in testicular tissue of heterozygous and homozygous bulls were observed (L475). Therefore, it follows an underlying liability model rather than a recessive model, where heterozygous individuals are expected to be similar to the wild type homozygotes.

The heritability for tail and head anomalies were zero or close to zero, though both had moderate repeatability estimates. This may be discussed.

L435: date -> data

L470: wich -> which

L555: Ø?? With such large variation in ejaculate number, the median may be given rather than mean.

L580: Replace ‘either semen quality parameter’ -> a semen quality phenotype.

L590: Please edit the sentence ‘Because the means … analyses’.

L635: (qchisq(1-p,1)’)’

L700: mt and wt are not defined yet.

L1115: (m) -> (mt)

**Have all data underlying the figures and results presented in the manuscript been provided?**

Reviewer #1: Yes

Reviewer #2: Yes

PLOS authors have the option to publish the peer review history of their article (what does this mean?). If published, this will include your full peer review and any attached files.

Reviewer #1: No

Reviewer #2: No

---

## [Decision Letter · Decision Letter 1]

21 Apr 2020

Dear Dr Pausch,

Thank you very much for submitting your Research Article entitled 'Activation of cryptic splicing in bovine WDR19 is associated with reduced semen quality and male fertility' to PLOS Genetics. Your manuscript has been re-evaluated by the two original reviewers. The reviewers appreciated the attention to an important topic but one point that needs addressing. As this is a text edit, this should be able to be taken care of quickly.

[LINK]

Yours sincerely,

Moira K. O'Bryan

Associate Editor

PLOS Genetics

Hua Tang

Section Editor: Natural Variation

PLOS Genetics

If the authors can rephrase the region of the text suggested by reviewer 1, the manuscript will be accepted.

Reviewer's Responses to Questions

**Comments to the Authors:**

Reviewer #1: In the revised version, the authors added an analysis of the important NSUN7 candidate gene concluding that this gene’s expression was intact in a homozygote for the male infertility BTA6 QTL. Although, the 5’ end of NSUN7 was under-expressed by this homozygote (Supporting File 8, SAMN14485268) comparing to the Angus controls (please correct to SAMN09205187, in panel B, Supporting File 8), I tend to accept their interpretation. Yet, I suggest the use of a more careful phrasing while describing this “intact” expression, introducing explanation why it was lower for the 5’ end comparing to the controls. Generally, the authors addressed my concerns and therefore I think that their work would be a valuable contribution to Plos Genetics.

Reviewer #2: The authors have revised the manuscript satisfactorily. I have no further comments.

**Have all data underlying the figures and results presented in the manuscript been provided?**

Reviewer #1: Yes

Reviewer #2: Yes

PLOS authors have the option to publish the peer review history of their article (what does this mean?). If published, this will include your full peer review and any attached files.

Reviewer #1: No

Reviewer #2: No

---

## [Editor Report · Decision Letter 2]

28 Apr 2020

Dear Dr Pausch,

We are pleased to inform you that your manuscript entitled "Activation of cryptic splicing in bovine WDR19 is associated with reduced semen quality and male fertility" has been editorially accepted for publication in PLOS Genetics. Congratulations!

Yours sincerely,

Moira K. O'Bryan

Associate Editor

PLOS Genetics

Hua Tang

Section Editor: Natural Variation

PLOS Genetics

Comments from the reviewers (if applicable):

This manuscript is now ready for publication. Congratulations.

**Data Deposition**

http://datadryad.org/submit?journalID=pgenetics&manu=PGENETICS-D-20-00128R2

**Press Queries**

---

## [Editor Report · Acceptance letter]

6 May 2020

PGENETICS-D-20-00128R2 

Activation of cryptic splicing in bovine WDR19 is associated with reduced semen quality and male fertility 

Dear Dr Pausch, 

We are pleased to inform you that your manuscript entitled "Activation of cryptic splicing in bovine WDR19 is associated with reduced semen quality and male fertility" has been formally accepted for publication in PLOS Genetics! Your manuscript is now with our production department and you will be notified of the publication date in due course.

With kind regards,

Jason Norris

PLOS Genetics

On behalf of:
